# Purple Corn Extract Improves Dry Eye Symptoms in Models Induced by Desiccating Stress and Extraorbital Lacrimal Gland Excision

**DOI:** 10.3390/nu15245063

**Published:** 2023-12-11

**Authors:** Jae-Min Lee, Arin Choi, Hee-Hwan Lee, Sang Jae Park, Byung-Hak Kim

**Affiliations:** MEDIENCE Co., Ltd., Chuncheon 24232, Republic of Korea; jmlee@mdec.co.kr (J.-M.L.); archoi@mdec.co.kr (A.C.); hhlee@mdec.co.kr (H.-H.L.); psmarch@naver.com (S.J.P.)

**Keywords:** dry eye disease, purple corn extract, anthocyanin, cyanidin-3-*O*-glucoside, inflammation, regulated cell death

## Abstract

Dry eye disease (DED) occurs when there are not enough tears, and the associated symptoms—burns, itching, and a gritty feeling in the eye—can cause great discomfort. The purpose of this study was to evaluate the therapeutic effect of purple corn extract (PCE) on DED. Pretreatment with PCE prevented desiccation-stress-induced cell damage in human retinal pigment epithelial cells and primary human corneal epithelial cells. Furthermore, PCE reduced the mRNA expression of inflammatory mediators in the induction of desiccation stress. The therapeutic effects of PCE on DED were evaluated in an animal model with induced unilateral excision of the exorbital lacrimal gland. The administration of PCE was effective at recovering tear production, corneal surface irregularity, and conjunctival goblet cell density, as well as at reducing apoptotic cell death in the outer layer of the corneal epithelium. Collectively, PCE improved dry eye symptoms, and, therefore, it could be a potential agent to ameliorate and/or treat DED.

## 1. Introduction

Age-related dry eye disease (DED) is a multifactorial and common ocular surface disorder causing immense inconvenience, which adversely affects the quality of life of patients. With increasing longevity, this is a growing health problem globally, which affects vision in severe cases [1,2,3]. DED is primarily classified in two categories: aqueous-deficient DED with lack of tear production and evaporative DED with increased tear evaporation [1,4]. The ocular surface plays an essential role in maintaining moisture via the tear film, which is composed of several components, including water, lipids, mucins, cytokines, growth factors, and antimicrobial peptides [5,6]. On the ocular surface, the tear film lubricates and defends against pathogens. As we age, the tear film homeostasis changes, affecting the lacrimal glands, goblet cells, and meibomian glands in particular. These changes may lead to aqueous-deficient DED and further injury to the ocular surface through decreased secretion of tear volume [7,8,9,10].

Patients with DED may experience many uncomfortable symptoms, including photophobia, pain, burns, itching, and a gritty feeling in the eye. Although aging is a gradual and continuous process of natural change, it can be accompanied by numerous biological and physiological changes. Inflammaging, which is a chronic and low-grade inflammatory state, occurs during the aging process. It is characterized by an increase in the production of pro-inflammatory mediators. This condition accelerates tear instability, with tear production and DED symptoms, and the induction of stress-related signal transductions on the ocular surface [3,11,12]. Consequently, the regulation of inflammation can be a favorable therapeutic approach for DED. In fact, anti-inflammatory agents are used to reduce the signs and symptoms of DED and restore homeostasis in the tear film and ocular surface [12,13,14].

Originally, purple corn (*Zea mays* L.) came from the Andean Region of South America and was used to make traditional foods and beverages. It is now widely cultivated all over the world and has recently become an emerging star in the novel ingredients market and the pharmaceutical industry, as it includes a number of biologically active compounds, such as anthocyanins and other functional phytochemicals, that are beneficial for health [15,16]. Among the anthocyanins, cyanidin-3-*O*-glucoside (C3G), the 3-*O*-glycosylated form of cyanidin, is one of the most common rich compounds of purple corn. It has been reported to have several health-promoting effects, such as antioxidant, anti-inflammation, antidiabetic, antiangiogenetic, and anticarcinogenic properties [15,16,17,18]. Because of their high anthocyanin content, the husks and cobs of purple corn can be optional sources of anthocyanin for food supplements; however, they are commonly recognized as by-products and are largely discarded because of their low utilization. For this reason, it is essential to develop a number of products with benefits for health that can be obtained from the husks and cobs of purple corn. In the present study, we prepared purple corn extract (PCE) from the husks and cobs of purple corn and evaluated the biological activities of PCE on eye health, especially DED, in both in vitro and in vivo models.

## 2. Materials and Methods

### 2.1. Preparation of Purple Corn Extract

The hybrid grains of purple corn were developed and registered by Gangwon Agricultural Research and Extension Services (Chuncheon, Republic of Korea). As previously described, the PCE was prepared from the dried husks and cobs of the purple corn [19]. Briefly, equal amounts of husks and cobs were repeatedly extracted with a 10-fold volume of 30% ethanol for 6–12 h at 60 °C. The extract was subsequently filtered using Whatman Qualitative No. 1 filter paper, concentrated at 60 °C in a rotary evaporator (Eyela, Tokyo, Japan), and PCE powder was obtained by spray drying using a spray drier. The powder was dissolved in dimethyl sulfoxide (DMSO) as a stock solution at a concentration of 200 mg/mL, and in vitro biological assays were then carried out. The DMSO used as a vehicle had a final concentration of 0.15%, except for the cell viability assay.

### 2.2. High-Performance Liquid Chromatography (HPLC) Analysis of Cyanidin-3-O-Glucoside

Cyanidin-3-*O*-glucoside (C3G) is one of the major components of PCE. In order to quantify the amounts of C3G, the extracted PCE was analyzed using an LC-20AD pump (Kyoto, Japan). Quantitative analysis was carried out using a reverse-phase system, and separation was performed on a Unison US-C18 column (25 cm × 4.6 mm, 5 μm). The column temperature was maintained at 40 °C, and the flow rate was set to 1 mL/min. The mobile phase comprised 0.1% trifluoroacetic acid (TFA) in water and acetonitrile, and the elution was carried out at 85–15% for 40 min. The injection volume was 10 μL, and the UV absorbance was recorded at 525 nm. The standard reagent against C3G was obtained from ChromaDex (Los Angeles, CA, USA).

### 2.3. Cell Culture

The human retinal pigment epithelial cell line ARPE-19 (CRL-2302) and primary human corneal epithelial cells (HCEpiC, PCS-700-010) were obtained from the American Type Culture Collection (Manassas, VA, USA). The ARPE-19 cells were maintained in DMEM/F-12 (HyClone, Logan, UT, USA) and supplemented with 10% heat-inactivated fetal bovine serum (FBS, HyClone) and 1% penicillin/streptomycin (Welgene, Gyeongsan, Republic of Korea). According to the manufacturer’s instructions, primary HCEpiCs were maintained in corneal epithelial cell basal medium (PCS-700-030, ATCC) that contained growth supplements (PCS-700-040, ATCC).

### 2.4. Cell Viability Assay

Cells were seeded at a density of 2 × 10^4^ cells/well in 96-well plates containing fresh medium. When the cells reached their experimental time points after an overnight incubation, the cells were incubated with vehicle alone, with different concentrations of PCE, or with omega-3 for 24–72 h. The cell viability was determined at 450 nm using a microplate reader (Molecular Devices, Sunnyvale, CA, USA) after being further incubated for 2–4 h at 37 °C following the addition of EZ-CyTox Enhanced Cell Viability Assay Reagent (Daeil Lab Service, Seoul, Republic of Korea). In order to induce desiccating stress, the cells were exposed at various time points of the desiccation stress or exposed to desiccation stress for 30 min, followed by the removal of the cultured medium, and then the cell viability was determined using an EZ-CyTox Enhanced Cell Viability Assay Reagent. The crude source of omega-3 (F8020) used as a positive control was obtained from Sigma-Aldrich (St. Louis, MO, USA).

### 2.5. Quantitative Real-Time Polymerase Chain Reaction (qRT-PCR)

According to the manufacturer’s instructions, the total RNA was extracted using an AccuPrep Universal RNA Extraction Kit (Bioneer, Daejon, Republic of Korea), and the cDNA was synthesized using an WizScript cDNA Synthesis Kit (Wisbio Solutions, Seongnam, Republic of Korea). Using a Dyne Fast qPCR 2× PreMix (Dyne Bio, Seongnam, Republic of Korea) with the Mic qPCR Analysis Software v2.12.0 (Bio Molecular Systems, Upper Coomera, Australia), the qRT-PCR was performed. A comparative C_t_ quantification of the data was performed to derive a value for each sample, which was normalized to the value for the housekeeping glyceraldehyde-3-phosphate dehydrogenase (GAPDH) gene. The sequences of the primers are summarized in Table 1.

### 2.6. Animal Experiments

Six-week-old male Sprague–Dawley (SD) rats were obtained from KoaTech (Pyeongtaek, Republic of Korea), and these rats were acclimatized for 7 days. In order to induce experimental DED, the rats were deeply anesthetized with isoflurane (JW Pharmaceutical, Seoul, Republic of Korea), and their left exorbital lacrimal gland was surgically excised. The next day, the rats were randomly separated into five groups (*n* = 5) and orally administered either the vehicle (vehicle-treated group), PCE (30, 80, or 150 mg/kg, PCE-treated group), or omega-3 (100 mg/kg, omega-3-treated group) every day for 7 days. Sham-operated rats were assigned to the normal group. Prior to administration, the PCE and omega-3 were freshly prepared in distilled water every day. The experimental protocol was approved by the Institutional Animal Care and Use Committee (IACUC) of Chonbuk National University Hospital Non-Clinical Evaluation Center (IACUC approval No. 2022-42).

### 2.7. Tear Volume Measurement

Tear volume was assessed using the phenol red thread tear test (Zone Quick, FCI Ophthalmics, Pembroke, MA, USA) following treatment with PCE or omega-3 for 7 days. The cotton thread was placed in the lateral canthus for 30 s, and the length of the color-changed thread was measured. The tear volume was measured in both eyes.

### 2.8. Corneal Surface Irregularity

As previously mentioned, the change in the ocular surface induced by desiccating stress was determined [20]. The irregularity was measured by the reflected light from the fiber-optic ring illuminator (SZ51, Olympus, Tokyo, Japan) on the corneal surface. The values were scored as follows: 0, no distortion; 1, distortion in 1/4 of the reflected ring shape; 2, distortion in 2/4; 3, distortion in 3/4; 4, distortion in 4/4; and 5, severe distortion and no ring shape could be recognized.

### 2.9. Periodic Acid–Schiff (PAS) Staining

Five-micrometer-thick sections from the paraffin-embedded cornea specimens were mounted on glass slides, deparaffinized, hydrated with grade ethanol, and then oxidized in a 1% periodic acid solution for 5–10 min. These sections were rinsed with distilled water and reacted with Schiff’s reagent for 20–30 min. After being rinsed with tap water, the sections were dehydrated with grade ethanol, cleared with xylene, mounted in DPX (Sigma-Aldrich), and then observed under an inverted fluorescence microscope (Carl Zeiss, Jena, Germany). Counterstaining of the sections was performed using 0.2% Mayer’s Hematoxylin Solution (Sigma-Aldrich).

### 2.10. Terminal Deoxynucleotidyl Transferase-Mediated dUTP Nick End Labeling (TUNEL) Staining

According to the manufacturer’s instructions (In Situ Cell Death Detection Kit, Roche, Germany), the apoptotic cells on the corneal tissue were determined with TUNEL staining. The TUNEL-positive cells were counted using an inverted fluorescence microscope (Carl Zeiss).

### 2.11. Statistical Analysis

The results were derived from at least three independent experiments, and all data are presented as the mean ± standard error of mean (SEM). The statistical analysis was carried out with the use of GraphPad Prism 5.0 (GraphPad Software, San Diego, CA, USA), and the significance was determined using a two-tailed Student’s *t*-test. Differences were considered statistically significant at *p* < 0.05.

## 3. Results

### 3.1. HPLC Analysis of PCE

Purple corn contains a wide range of anthocyanins that are present in its husks, barks, stems, and kernels, with C3G being one of the most common constituents and responsible for its purple color [16,17,18,21,22]. Developed by Gangwon Agricultural Research and Extension Services, the PCE was prepared from a mixture of the dried husks and cobs of the purple corn. We obtained a standard compound of C3G and performed an HPLC analysis to determine the amount of C3G in the PCE. The data from the HPLC analyses of the C3G standard and the PCE used in the chromatograms were obtained by observing the detector responses at 525 nm. We observed two major peaks in the chromatogram of the PCE, one of which, C3G, was detected at about 7.8 min; its content was 18.43 ± 0.58 mg/g (Figure 1).

### 3.2. PCE Protects Cells against Damage from Desiccation Stress

The cell viability assay was performed in human retinal pigment epithelial and primary human corneal epithelial cells to determine the cytotoxicity of PCE. The cells were maintained for up to 72 h with various concentrations of PCE or omega-3. The PCE did not affect any cytotoxic activity up to 3000 μg/mL for 72 h in both cell lines (Figure 2A,B). However, in the cell lines with high concentrations above 1000 μg/mL, omega-3 enhanced the viability of cells (Figure 2C,D).

Desiccation stress was induced in ARPE-19 and HCEpiC cells at different time points to mimic DED in a cell culture system, and then the cell viability was observed. As previously reported, desiccation stress was induced through air exposure on a clean bench [23]. The cell viability slightly decreased at 20 min and dramatically decreased at 30 min (Figure 2E). Therefore, the time point for desiccation stress was set to 30 min for the in vitro experiments. To examine the protective effect of the PCE or omega-3 against cell damage induced by desiccation stress, ARPE-19 or HCEpiC cells were preincubated with different concentrations of PCE or omega-3 for 24 h prior to exposure to desiccation stress for 30 min. In the presence of PCE or omega-3, the cell viability was restored effectively in a concentration-dependent manner, and this indicates the protective effects of PCE or omega-3 against desiccation stress (Figure 2F). On the basis of the results of the cell viability assays, we used concentrations of up to 300 μg/mL of PCE or omega-3 in the in vitro experiments.

### 3.3. PCE Decreases the mRNA Level of Inflammatory Mediators

Inflammation is known to further accelerate DED and is an important DED biomarker that is considered in the regulated cell death (RCD) of dry eye etiopathology [24]. Interleukin-6 (IL-6) and tumor necrosis factor-alpha (TNF-α), in particular, are correlated with necroptosis, and IL-1β is associated with pyroptosis [24]. To confirm whether the cellular cytoprotective effect of PCE is due to its anti-inflammatory effect, the mRNA expression of inflammatory mediators was determined with qRT-PCR. The mRNA expression of inflammatory molecules, which include IL-1β, IL-6, IL-8, IL-12, TNF-α, inducible nitric oxide synthase (iNOS), and cyclooxygenase-2 (COX-2), was increased approximately 2–4-fold that of the normal conditions via the induction of desiccation stress in the ARPE-19 and HCEpiC cells. This demonstrates that under desiccation conditions, the inflammatory response is activated. In a concentration-dependent manner, the PCE caused an effective decrease in their mRNA expression, and its effects were comparable to or stronger than those of omega-3 (Figure 3 and Figure 4). These findings demonstrate that PCE has anti-inflammatory effects in dry eye syndromes, and this activity may play an essential role in its cytoprotective effect.

### 3.4. PCE Improves Tear Production and Corneal Irregularity in Rats with DED

An in vivo model of DED was developed by surgical excision of the unilateral exorbital lacrimal gland in SD rats to evaluate the effect of PCE on eye health. PCE was administered orally at a constant concentration every day for 7 days, and then the tear volume was measured using a phenol-red-impregnated cotton thread. Tear production was reduced compared with the normal group for rats that were experimentally exposed to a DED model, suggesting that the DED model was well established. PCE administration effectively recovered the tear volume in a concentration-dependent manner. At high concentrations, the effect was almost normal and stronger than that of omega-3 (Figure 5A).

DED is associated with increased frequent blinking without the production of tears, causing damage to the cornea surface due to scratching. These phenomena alter the corneal epithelial barrier function and tear film integrity and cause inflammation. Therefore, the extent of corneal damage was determined by taking pictures of the rat’s eye, which reflected a white ring of light. The DED model group exhibited irregular circle shapes of light; however, the administration of PCE significantly restored these shapes to those of the normal group, and the effect was stronger than that of omega-3 (Figure 5B,C). These findings suggest that PCE may possibly improve eye health, especially DED, by maintaining moisture in the eye.

### 3.5. PCE Increases Corneal Goblet Cell Density on the Ocular Surface

Corneal mucins are essential to defend and preserve the integrity of the ocular surface, and alterations in corneal mucins have been found to be hallmarks of DED. Specifically, a decrease in goblet cell density is associated with the seriousness of DED, and this decrease is accompanied by a decrease in mucin production [25,26]. Therefore, the density of the goblet cells was analyzed for an indirect assessment of mucin production using immunohistochemistry for PAS staining, which can selectively stain acidic mucus, a mucin component contained in goblet cells, with red color. The goblet cell density remarkably decreased in the rats in the DED model group compared to those in the normal group, but it effectively increased in a concentration-dependent manner with the administration of PCE, and its effect was comparable to or stronger than that of omega-3 (Figure 6). The protective effect of PCE on the goblet cell density clearly indicates that PCE probably protects corneal surface damage and maintains tear film integrity by increasing the production of mucin, thereby improving dry eye symptoms.

### 3.6. PCE Prevents Apoptotic Cell Death in the Corneal Epithelium

DED is mainly linked to a lack of moisture in the eye, which leads to inflammation resulting in cell death associated with stress [24,27]. Recently, targeting RCD has been an emerging field and innovative strategy in the therapeutics of DED and ocular surface dysfunction. Therefore, protecting cell death from cell damage caused by stress is an important modulator of DED-related eye and ocular health improvement. In order to determine whether PCE has cytoprotective effects towards an in vivo model of DED, TUNEL staining was conducted in the tissue sections of the cornea of rats to detect dead cells by mediated apoptosis. TUNEL-positive cells were dramatically increased in the corneal epithelium and lacrimal glands of the DED rats compared to those in the normal control rats. This indicates that RCD, particularly apoptotic cell death, increased. However, this increase in apoptotic cell death was effectively reduced by PCE administration, and this effect was comparable to or stronger than that of omega-3 (Figure 7).

## 4. Discussion

DED, one of the most prevalent eye syndromes, affects millions of people around the world and becomes more common as people age. Recently, the incidence rate has rapidly been increasing in the younger population due to frequent use of mobile phones and video screens, use of contact lenses, and environmental factors, such as air pollution and low humidity [2,28]. These factors decrease tears in the eyes, leading to hyperosmolarity of tear film and inflammation of the ocular surface that causes damage to the ocular surface. Tear production deficiency is associated with various pathophysiological mechanisms such as decreasing cell volume, dysfunction of the DNA repair systems, and promotion of reactive oxygen species (ROS) generation and apoptotic cell death in ocular surface cells [20,29]. Therefore, care and attention should be paid to eye health, not only for middle-aged people but also for early age people.

Artificial tears, lubricants, and various ophthalmic drops, such as anti-inflammatory drugs, immunosuppressants, and steroids, are currently used to treat DED, but these therapeutics concentrate on relieving clinical symptoms rather than unraveling the fundamental causes of DED [30]. These limitations indicate the need to develop novel therapeutic agents to treat DED. We investigated the protective effects of purple corn on DED models for the improvement of eye health by treating the fundamental causes of DED. Purple corn is a natural product, which is newly developed and registered by Gangwon Agricultural Research and Extension Services, and the extract, PCE, was prepared from a mixture of the husks and cobs of purple corn [19]. To examine the potential protective effects of PCE against DED, the DED models were constructed with desiccation stress induced by air exposure on a clean bench in vitro and by surgical excision of the unilateral exorbital lacrimal gland in SD rats.

Purple corn contains various biologically functional components such as anthocyanins, phenolic acids, and flavonoids. These components play an important role in its health benefits, which include antioxidant, anti-inflammatory, antimutagenic, anticarcinogenic, anticancer, and antiangiogenetic properties; blood pressure regulation; cardiovascular health benefits; and improvement in lifestyle diseases, like obesity, diabetes, hyperglycemia, and other linked diseases [15,16,17,18]. On the basis of our findings, PCE effectively improved eye health in both the in vitro and in vivo DED models. PCE exhibited cytoprotective and anti-inflammatory activities in ARPE-19 and HCEpiC cells induced by desiccation stress in the in vitro experiments. Specifically, the anti-inflammatory activity of PCE may be closely associated with cytoprotective effects, because inflammatory molecules promote RCD in DED conditions during inflammaging [3,11,12,24]. Particularly, inflammatory cytokines, such as IL-1β, IL-6, IL-17A, IL-18, and TNF-α, are associated with RCDs, such as apoptosis, necroptosis, and pyroptosis [24]. Using TUNEL staining, the cytoprotective effect of PCE was further validated in the in vivo model of DED. PCE administered orally significantly reduced TUNEL-positive cells in the corneal epithelium and lacrimal glands of the DED rats, which indicates the anti-apoptotic activity of PCE. It is thus predicted that the cytoprotective effects of PCE have a strong correlation with its anti-inflammatory activity.

Anthocyanins, which are a class of water-soluble phenolic compounds with a content in the range of 6.8–82.3 mg/g, are mainly responsible for the dark purple–red color of purple corn. The anthocyanin content in purple corn is much higher than that in other products known to be rich in anthocyanin, such as red grapes, blueberries, and aronia [21,31,32]. The major anthocyanins in purple corn are C3G, pelargonidin-3-*O*-glucoside, peonidin-3-*O*-glucoside, cyanindin-3-*O*-(6″-malonylglucoside), pelargonidin-3-*O*-(6″-malonylglucoside), and peonidin-3-*O*-(6″-malonylglucoside) [21,33,34,35,36,37]. It is well known that these components play an essential role in beneficial biological activities. In the HPLC analysis of the PCE, two major peaks and three minor peaks were observed; one of the major peaks was identified as C3G, and its content was 18.43 ± 0.58 mg/g. Interestingly, Wistar rats and F344 rats in acute and 90-day subchronic oral toxicity studies showed no toxicity symptoms at high dosages of purple corn pigment, suggesting that the products made by purple corn are safe [38,39]. According to the results reported by Nabae and colleagues, the no-observed-adverse-effect level (NOAEL) values were determined to be 3542 and 3849 mg/kg body weight/day in male and female F344 rats, respectively [39]. To date, PCE has been found to be safe enough that there is no limitation to the recommended dosage of the consumption of purple corn anthocyanin for human health.

According to the Tear Film & Ocular Surface Society (TFOS) Dry Eye Workshop II (TFOS DEWS II) in 2017, dry eye is a multifactorial disease of the ocular surface characterized by a loss of homeostasis of the tear film and is accompanied by ocular symptoms, in which tear film instability and hyperosmolarity, ocular surface inflammation and damage, and neurosensory abnormalities play etiological roles [40]. Based on these definitions, DED is a disease described by the disruption of homeostasis, whether in the tear film, anatomy, or the nervous system. Taken together, the main basis of DED is an insufficient amount of tears resulting in various uncomfortable problems on the ocular surface. In the DED model rats, our in vivo results demonstrated that the administration of PCE restored tear production and corneal irregularity, as well as increased the corneal goblet cell density. These findings indicate that PCE can improve eye health in DED by restoring the normal homeostasis of tears and improving the intrinsic properties of the tear film and the ocular surface.

## 5. Conclusions

In conclusion, PCE exhibited cytoprotective and anti-inflammatory effects in dry conditions induced by desiccation stress. Moreover, treatment with PCE in the animal model was effective at restoring the progression of DED by improving tear production, corneal irregularity, and goblet cell density. Our findings, thus, indicate that PCE can be used as a healthy functional food material for eye health, particularly in DED cases.

## Figures and Tables

**Figure 1 nutrients-15-05063-f001:**
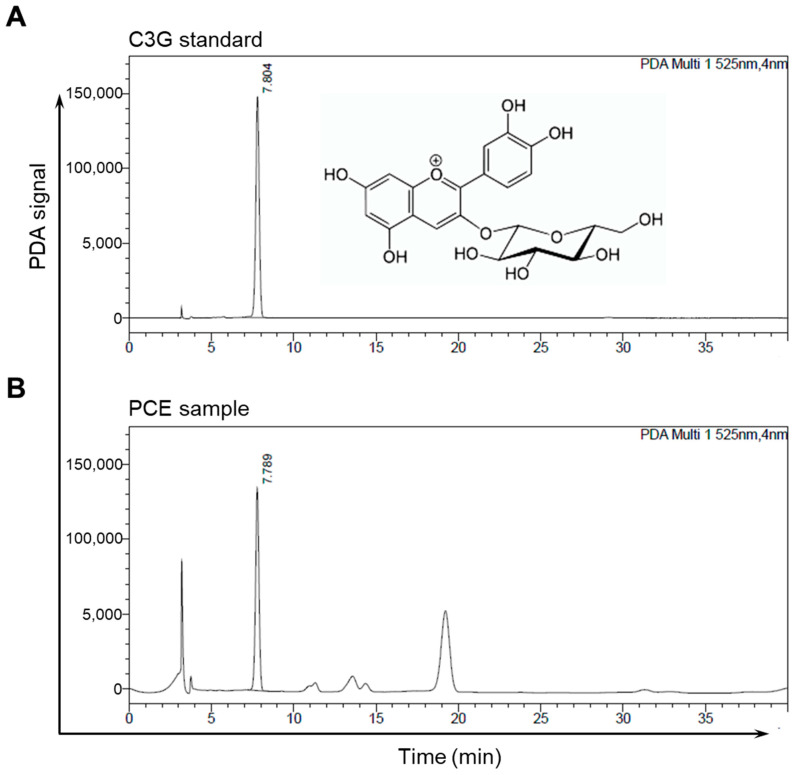
Representative chromatographs of HPLC: (**A**,**B**) HPLC chromatograms of the C3G standard (**A**) and PCE (**B**) were detected at 525 nm. The peak of C3G appeared at a retention time of 7.8 min.

**Figure 2 nutrients-15-05063-f002:**
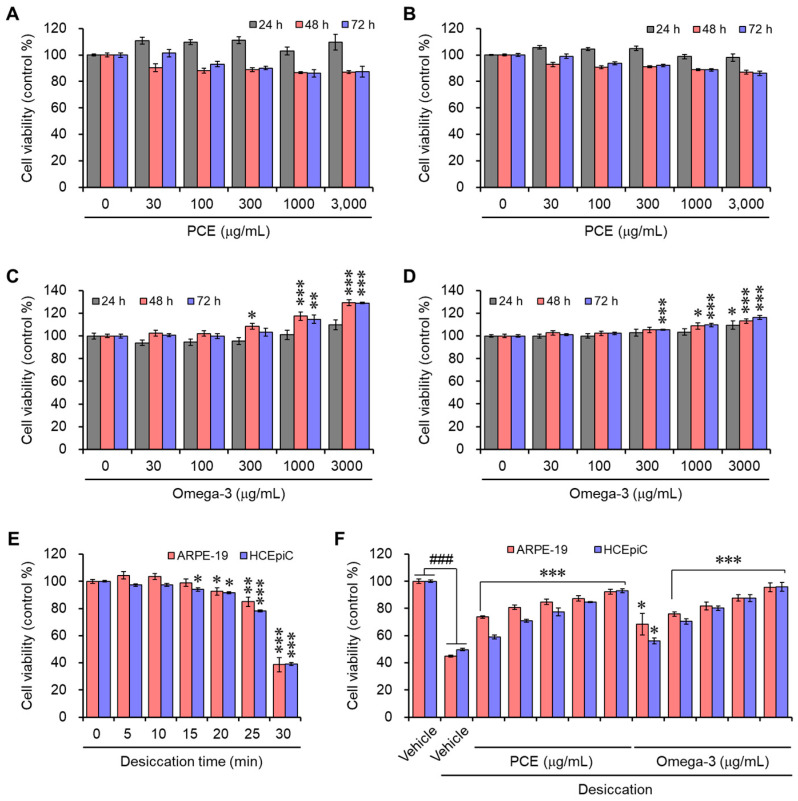
Effects of PCE and omega-3 on the viability of ARPE-19 and HCEpiCs: (**A**–**D**) ARPE-19 (**A**,**C**) or HCEpiCs (**B**,**D**) were incubated with different concentrations of C3G (**A**,**B**) or omega-3 (**C**,**D**) for the indicated time points, and their viability was determined. (**E**) ARPE-19 or HCEpiCs were exposed to different time points of air to induce desiccation stress, and their viability was determined. In all cases, the viability is expressed as a % of the control. The results are presented as the mean ± SEM (*n* = 3). * *p* < 0.05, ** *p* < 0.005, and *** *p* < 0.0001 versus the vehicle-treated group. (**F**) ARPE-19 or HCEpiCs were preincubated with different concentrations of C3G or omega-3 for 24 h; then, they were exposed to desiccation stress for 30 min, and the cell viability was determined. In all cases, the viability is expressed as a % of the control. The results are presented as the mean ± SEM (*n* = 3). ^###^ *p* < 0.0001 versus the vehicle-treated group. * *p* < 0.05 and *** *p* < 0.0001 versus the desiccation-stress-induced vehicle group.

**Figure 3 nutrients-15-05063-f003:**
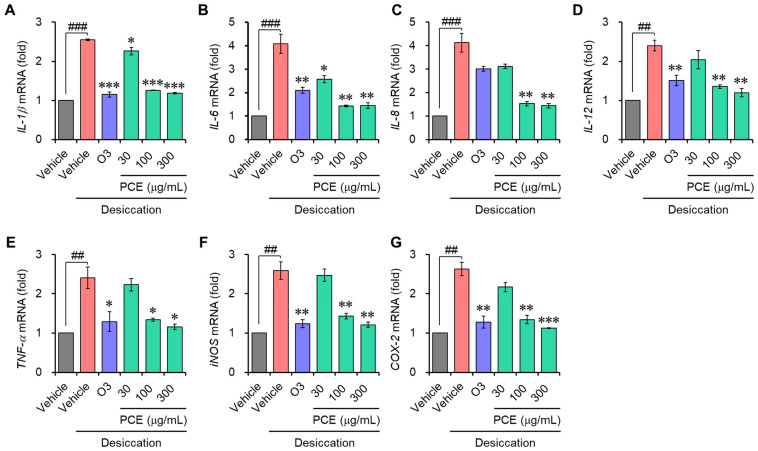
Effects of PCE on mRNA expression of inflammatory mediators in ARPE-19: (**A**–**G**) ARPE-19 cells were preincubated with different concentrations of PCE for 2 h, exposed to desiccation stress for 30 min, and further incubated for 6 h. The total RNA was extracted, and the mRNA expression of inflammatory mediators was analyzed using real-time PCR. The results are expressed as relative values compared to the vehicle-treated control and presented as the mean ± SEM (*n* = 3). ^##^ *p* < 0.005 and ^###^ *p* < 0.0001 versus the vehicle-treated group. * *p* < 0.05, ** *p* < 0.005, and *** *p* < 0.0001 versus the desiccation-stress-induced vehicle group. O3, omega-3, 300 μg/mL.

**Figure 4 nutrients-15-05063-f004:**
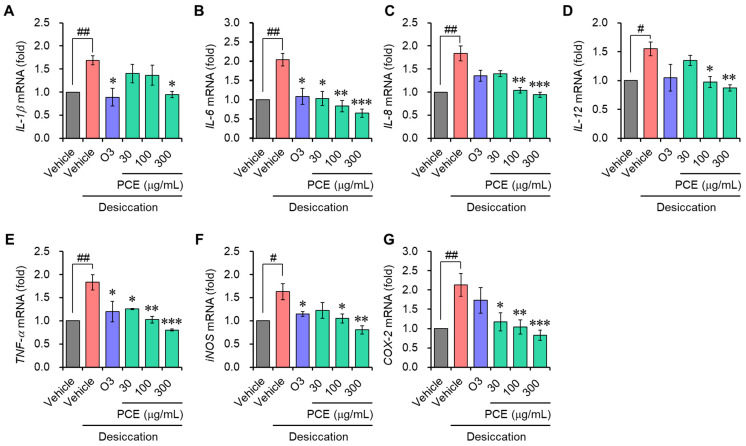
Effects of PCE on the mRNA expression of inflammatory mediators in HCEpiCs: (**A**–**G**) HCEpiCs were preincubated for 2 h with different concentrations of PCE, exposed to desiccation stress for 30 min, and further incubated for 6 h. The total RNA was extracted, and the mRNA expression of the inflammatory mediators was analyzed using real-time PCR. The results are expressed as relative values compared to the vehicle-treated control and presented as the mean ± SEM (*n* = 3). ^#^ *p* < 0.05 and ^##^ *p* < 0.005 versus the vehicle-treated group. * *p* < 0.05, ** *p* < 0.005, and *** *p* < 0.0001 versus the desiccation-stress-induced vehicle group. O3, omega-3, 300 μg/mL.

**Figure 5 nutrients-15-05063-f005:**
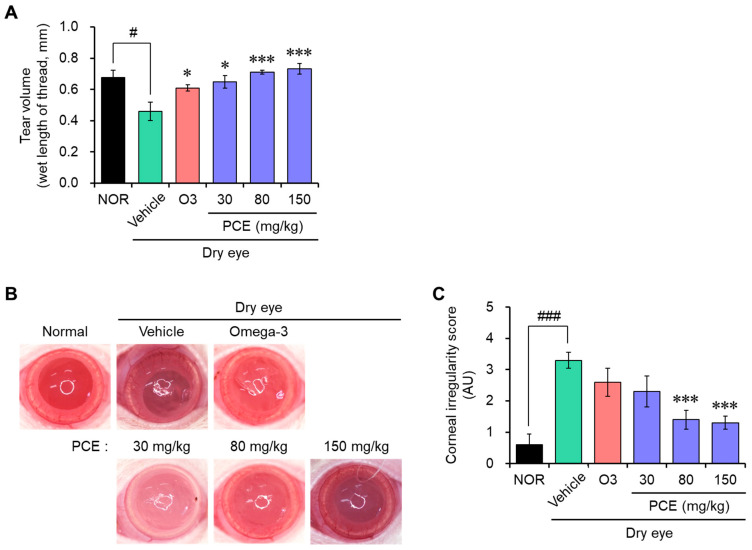
Effects of PCE on tear secretion and corneal irregularity in DED model rats. (**A**) Tear volume was measured using a phenol red thread test. Tear volume is expressed in milliliters of thread wetted by the tear and turned red in color. (**B**,**C**) Reflected images (**B**) and the scores (**C**) of a white ring from the fiber-optic ring illuminator of a stereomicroscope, and the corneal irregularity was graded based on the number of distorted quarters in the reflected white ring, as described in Section 2.8. The results are presented as the mean ± SEM (*n* = 5). ^#^
*p* < 0.05 and ^###^
*p* < 0.0001 versus the normal group. * *p* < 0.05 and *** *p* < 0.0001 versus the DED-induced group. O3, omega-3, 100 mg/kg.

**Figure 6 nutrients-15-05063-f006:**
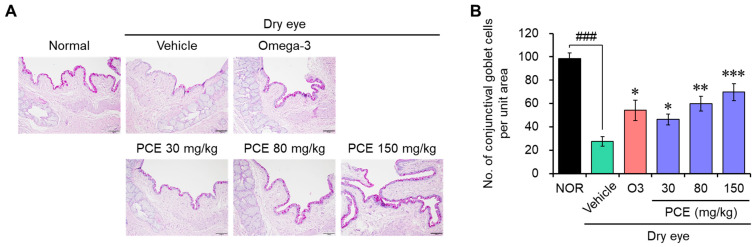
Effects of PCE on goblet cell density in DED model rats: (**A**,**B**) immunohistochemical analysis of the PAS staining was carried out to analyze the acidic mucus in goblet cells, and representative images (**A**) and PAS-positive cells in the corneal sections are shown as the conjunctival goblet cell density (**B**). The results are presented as the mean ± SEM (*n* = 5). ^###^
*p* < 0.0001 versus the normal group. * *p* < 0.05, ** *p* < 0.005, and *** *p* < 0.0001 versus the DED-induced group. O3, omega-3, 100 mg/kg.

**Figure 7 nutrients-15-05063-f007:**
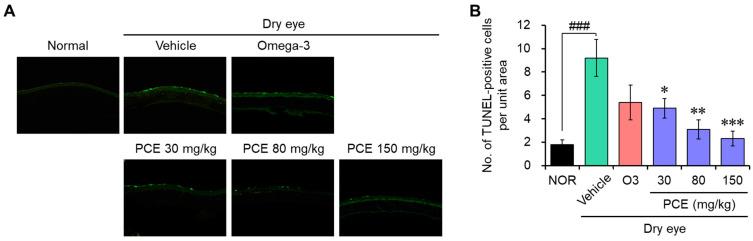
Effects of PCE on apoptotic cell death in the corneal epithelium of DED model rats: (**A**,**B**) TUNEL staining was carried out to analyze apoptotic cell death in the corneal epithelium, and representative images (**A**) and TUNEL-positive cells in the corneal sections (**B**) are shown. The results are presented as the mean ± SEM (*n* = 5). ^###^ *p* < 0.0001 versus the normal group. * *p* < 0.05, ** *p* < 0.005, and *** *p* < 0.0001 versus the DED-induced group. O3, omega-3, 100 mg/kg.

**Table 1 nutrients-15-05063-t001:** Primer sequences used for the quantitative real-time PCR.

Target Genes		Primer Sequences
*IL-1β*	Forward	5′-GCACGATGCACCTGTACGAT-3′
	Reverse	5′-AGACATCACCAAGCTTTTTTGCT-3′
*IL-6*	Forward	5′-CAGGAATTGAATGGGTTTGC-3′
	Reverse	5′-AAACCAAGGCACAGTGGAAC-3′
*IL-8*	Forward	5′-TTTTGCCAAGGAGTGCTAAAGA-3′
	Reverse	5′-AACCCTCTGCACCCAGTTTTC-3′
*IL-12*	Forward	5′-CTTGTGGCTACCCTGGTCCT-3′
	Reverse	5′-GAGTTTGTCTGGCCTTCTGG-3′
*TNF-α*	Forward	5′-CTGGGCAGGTCTACTTTGGG-3′
	Reverse	5′-CTGGAGGCCCCAGTTTGAAT-3′
*iNOS*	Forward	5′-GGTGGAAGCGGTAACAAAGG-3′
	Reverse	5′-TGCTTGGTGGCGAAGATGA-3′
*COX-2*	Forward	5′-GCCAAGCACTTTTGGTGGAG-3′
	Reverse	5′-GGGACAGCCCTTCACGTTAT-3′
*GAPDH*	Forward	5′-GACCACAGTCCATGCCATCA-3′
	Reverse	5′-TCCACCACCCTGTTGCTGTA-3′

## Data Availability

The data are not publicly available due to privacy and ethical restrictions.

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
