# Peer review of "Purple Corn Extract Improves Dry Eye Symptoms in Models Induced by Desiccating Stress and Extraorbital Lacrimal Gland Excision"

_nutrients, 2023, doi:10.3390/nu15245063_

Round 1

Reviewer 1 Report

Comments and Suggestions for Authors

In this research paper, the authors confirmed for the first time through in vitro and in vivo experiments that purple corn extract (PCE) can significantly improve dry eye disease (DED) symptoms: it could not only prevent the damages of human retinal pigment epithelial cells and primary human corneal epithelial cells caused by desiccation stress, but also effectively reduce the inflammatory response caused by the desiccation stress. Therefore, the authors believe that PCE could serve as a potential natural medicine for the treatment of DED.

This work helps us to understand the mechanism of PCE in treating DED and provides a potential drug option for DED patients.

Figure 3 and Figure 4:

It would be easier to understand if the meanings/representations of the black and pink bars are marked.

Line 288:” protecting cell death” to protecting from cell death?

Author Response

Comments and Suggestions for Authors

In this research paper, the authors confirmed for the first time through in vitro and in vivo experiments that purple corn extract (PCE) can significantly improve dry eye disease (DED) symptoms: it could not only prevent the damages of human retinal pigment epithelial cells and primary human corneal epithelial cells caused by desiccation stress, but also effectively reduce the inflammatory response caused by the desiccation stress. Therefore, the authors believe that PCE could serve as a potential natural medicine for the treatment of DED.

This work helps us to understand the mechanism of PCE in treating DED and provides a potential drug option for DED patients.

Figure 3 and Figure 4: It would be easier to understand if the meanings/representations of the black and pink bars are marked.

Response: We appreciate the Reviewer’s consideration and insightful comments with suggestions. In response to the Reviewer’s comments, we replaced Figure 3 and Figure 4 with new ones for easier understanding of the black and pink bars. In the Figures, the 0 labeled bars were replaced by a vehicle, and the vehicle was described in each Figure Legends and Materials and Methods.

Line 288:” protecting cell death” to protecting from cell death?

Response: In general, cell death is considered one of the most important mechanisms to maintain cell homeostasis and integrity of multicellular organisms. During DED, oxidative stress and inflammation occur and these conditions result in membrane disruption, damage to the corneal and conjunctival cells and ultimately regulated cell death such as apoptosis, ferroptosis, necrosis and pyroptosis. Therefore, targeting regulated cell death is one of the innovative strategies in the therapeutics of DED and ocular surface dysfunction. As the Reviewer’s commented, protecting cell death means protecting regulated cell death from cell damage caused by stress conditions, especially oxidative stress and inflammation during DED.

Reviewer 2 Report

Comments and Suggestions for Authors

This manuscript describes the beneficial effects on eye health, especially on age-related dry eye disease (DED), of a hydroethanolic extract of purple corn (PCE) obtained from the husks and cobs of those by-products. The effects have been clearly observed using both in vitro (two human epithelial cell lines one retinal pigment epithelial line and one corneal epithelial and in vivo models) and in vivo (Sprague Dawley rats with the left Ex orbital lacrimal gland surgically excised). The article contains clear results, and the discussion and conclusion are consistent with the results. References are also satisfactory.

 The following minor points would be addressed before definitive acceptance.

Materials & methods: The PCE powder was dissolved in dimethyl sulfoxide (DMSO) as a stock solution, 200 mg/mL. Later, cell cultures were treated with PCE concentrations up to 3 mg/ml (3000 mg/mL) and rats were treated orally with either vehicle, PCE (30, 80, or 150 mg/kg) every day for 7 days. Therefore, DMSO stock solution would be diluted. The solvent for those dilutions should be mentioned, and the amount of DMSO remaining would be calculated. In turn, a control using the maximal amount of DMSO would be conducted.

Line 104: Data about the composition of "crude source" omega3 would be also given. In turn, authors should give some references about the protective effect of those fatty acids against desiccation stress in ocular cells accounting for this choice as a positive control.

 Line 114: The letter size of GADPH should be repaired.

 Line 125: Please, clarify the five groups.

Line 148: Separate Terminaldeoxynucleotidyltransferase in two words

 Figure 2 and 3 and 4: The meaning of 0 and 0 labeled bars on the left of figure 2F (labeled as ###) would be clarified at the figure 2 footnote. What does vehicle-treated group mean? Is that pair (0, 0) in cell culture experiments comparable to the NOR and 0 labeled bars in Figures 5 to 7 (in vivo experiments)?

 Line 347-349: A Control to pure cyanidin-3-O-glucoside (C3G) would be recommendable to know if other PCE components (flavonoids or other anthocyanins described as lines 347-349) would have stronger effect that just C3G. In this manuscript, the effect is attributed from the beginning to C3G as a major component of purple corn extract, but Figure 1 indicates that other major component (retention time 19 min) and three minor components (retentions times 10-15 min) are also present in the PCE.

 Line 350: “knonw” would be repaired.

Comments on the Quality of English Language

Minor hypos. See comments to authors.

Author Response

Comments and Suggestions for Authors

This manuscript describes the beneficial effects on eye health, especially on age-related dry eye disease (DED), of a hydroethanolic extract of purple corn (PCE) obtained from the husks and cobs of those by-products. The effects have been clearly observed using both in vitro (two human epithelial cell lines one retinal pigment epithelial line and one corneal epithelial and in vivo models) and in vivo (Sprague Dawley rats with the left Ex orbital lacrimal gland surgically excised). The article contains clear results, and the discussion and conclusion are consistent with the results. References are also satisfactory.

 The following minor points would be addressed before definitive acceptance.

Materials & methods: The PCE powder was dissolved in dimethyl sulfoxide (DMSO) as a stock solution, 200 mg/mL. Later, cell cultures were treated with PCE concentrations up to 3 mg/ml (3000 mg/mL) and rats were treated orally with either vehicle, PCE (30, 80, or 150 mg/kg) every day for 7 days. Therefore, DMSO stock solution would be diluted. The solvent for those dilutions should be mentioned, and the amount of DMSO remaining would be calculated. In turn, a control using the maximal amount of DMSO would be conducted.

Response: We appreciate the Reviewer’s consideration and helpful comments with detailed suggestions. If fact, DMSO is an effective solvent and cytoprotectant agent that can induce diverse actions in experimental settings, ranging from metabolic stress to cytotoxic effects depending on the concentration used. Thus, it has been used as a preferred solvent for the dissolution of various compounds including pharmacological and biochemical reagents, in particular small hydrophobic molecules for in vitro and in vivo experiments. DMSO is considered to be used at lower concentrations because of its cytotoxic activity. The concentrations applied in cellular experiments are commonly known to have no cytotoxic effect in the range of 0.1 to 0.5%.

For in vitro assays, we prepared 200 mg/ml of PCE in DMSO as a stock solution and the final concentration was 0.15% to exclude cytotoxic effect, except for cell viability assay. In accordance with the Reviewer’s comments, we have referred the DMSO concentration in Materials and Methods (see lines 94-95). For in vivo assay, PCE was freshly prepared in distilled water every day (see lines 159 and165), therefore, DMSO is not relevant for in vivo experiments.

Line 104: Data about the composition of "crude source" omega3 would be also given. In turn, authors should give some references about the protective effect of those fatty acids against desiccation stress in ocular cells accounting for this choice as a positive control.

Response: The omega-3 was obtained from Sigma-Aldrich (catalog number, F8020) and referred to its catalog number in the manuscript. According to the datasheet, it contains palmitic plus stearic acid (total): ≤ 30.0% and 20.0 to 31.0% omega-3 (octadecatetraenoic, eicosapentaenoic and docosahexaenoic) fatty acids as triglycerides.

The effect of omega-3 on DED and ocular surface disease is well-known in a number of reports. The representative recent reports are represented below as the references.

  1. Giannaccare G, Pellegrini M, Sebastiani S, Bernabei F, Roda M, Taroni L, Versura P, Campos EC. Efficacy of Omega-3 Fatty Acid Supplementation for Treatment of Dry Eye Disease: A Meta-Analysis of Randomized Clinical Trials. 2019;38:565-573.
  2. O'Byrne C, O'Keeffe M. Omega-3 fatty acids in the management of dry eye disease-An updated systematic review and meta-analysis. Acta Ophthalmol. doi: 10.1111/aos.15255.
  3. Paik B, Tong L. Topical Omega-3 Fatty Acids Eyedrops in the Treatment of Dry Eye and Ocular Surface Disease: A Systematic Review. Int J Mol Sci. 2022;23:13156.
  4. Bhargava R, Pandey K, Ranjan S, Mehta B, Malik A. Omega-3 fatty acids supplements for dry eye - Are they effective or ineffective? Indian J Ophthalmol. 2023;71:1619-1625.
  5. Wang WX, Ko ML. Efficacy of Omega-3 Intake in Managing Dry Eye Disease: A Systematic Review and Meta-Analysis of Randomized Controlled Trials. J Clin Med.;12:7026.

Line 114: The letter size of GADPH should be repaired.

Response: According to the Gene/Protein Nomenclature Guidelines, human, mouse, rat, fish, worm, or fly (Drosophila) are named differently. In this study, GAPDH is a human-originated gene, and the letter size has been repaired by the Reviewer’s suggestion (lines 148-149).

Line 125: Please, clarify the five groups.

Response: According to the Reviewer’s suggestion, the five groups were clarified in the manuscript. The five groups are divided into vehicle-treated group, 30 mg/kg PCE-treated group, 80 mg/kg PCE-treated group, 150 mg/kg PCE-treated group, and 100 mg/kg omega-3-treated group. Distilled water was used as a vehicle because PCE and omega-3 samples were prepared from distilled water (see lines 156-158).

Line 148: Separate Terminaldeoxynucleotidyltransferase in two words

Response: According to the Reviewer’s suggestion, it was separated in the manuscript.

Figure 2 and 3 and 4: The meaning of 0 and 0 labeled bars on the left of figure 2F (labeled as ###) would be clarified at the figure 2 footnote. What does vehicle-treated group mean? Is that pair (0, 0) in cell culture experiments comparable to the NOR and 0 labeled bars in Figures 5 to 7 (in vivo experiments)?

Response: According to the Reviewer’s suggestion, as well as the Reviewer #1’s suggestion, the Figure 2F and Figures 3-7 were replaced with new ones to make the meaning more clearly. In the figures, the 0 was replaced by a vehicle and mentioned in the Figure Legend.

Line 347-349: A Control to pure cyanidin-3-O-glucoside (C3G) would be recommendable to know if other PCE components (flavonoids or other anthocyanins described as lines 347-349) would have stronger effect that just C3G. In this manuscript, the effect is attributed from the beginning to C3G as a major component of purple corn extract, but Figure 1 indicates that other major component (retention time 19 min) and three minor components (retentions times 10-15 min) are also present in the PCE.

Response: As with the Reviewer’s comments, PCE contains a variety of components such as anthocyanins and other functional phytochemicals, and the components exert many biological activities for health benefits. Although 6 major components of anthocyanins were reported from purple corn extract, 2 major components including C3G and three minor components including peonidin-3-O-glucoside were identified in the HPLC chromatograms of our prepared PCE. Unfortunately, we don't yet know the components of one of the major peaks and two of the minor ones. We agreed with the Reviewer’s comments, and have discussed C3G primarily, although all the functional phytochemicals are effective in biological activities.

Line 350: “knonw” would be repaired.

Response: According to the Reviewer’s suggestion, the typo was corrected.
